# Consistent microbial insights across sequencing methods in soil studies: the role of reference taxonomies

Niranjana Rose Edwin,[1,2,3] Aoife Duff,[4] Coline Deveautour,[5] Fiona Brennan,[3,4] Florence Abram,[2] Orla O'Sullivan[1,3]

**ABSTRACT** Microbes play an important role in soil functioning, underpinning food production systems and delivering an array of essential ecosystem services. To elucidate how these microbes relate to ecosystem functions, accurate identification and classification of soil microorganisms are important. We evaluated the comparability of shotgun and amplicon sequencing approaches by profiling soil microbiota from 131 diverse temperate grassland soils across Ireland. We assessed method comparability in terms of (i) detection and classification of the most abundant phyla, (ii) their capacity to differentiate samples based on their microbial community, and (iii) their capacity to link microbial communities to measured nitrogen cycle functions. Our findings reveal that both methods offer moderately similar outcomes, providing consistent detection of major phyla, similar microbial community differentiation patterns, and largely identifying the same relationships between the phyla and nitrogen functions. The variations observed between the two methods were mostly associated with differences in the choice of reference taxonomy. Amplicon sequencing represents a cost-effective, less computationally demanding option, while shotgun sequencing provides deeper taxonomic resolution and access to the latest databases, making it suitable for detailed microbial profiling. Our study underscores the need for careful method selection based on project requirements, database availability, and financial resources.

**IMPORTANCE** Studying the microorganisms in soil remains a challenge as soils are one of the most complex and diverse environments. Compounding these challenges is the lack of culturable representatives in soil, with over 99% of soil microorganisms yet to be cultivated in a laboratory setting. Leveraging next-generation sequencing technologies, which bypass traditional culture-dependent methods, scientists are now able to attain low-cost, high-throughput DNA sequencing that can detect even the rarest microorganisms within samples. The present study rigorously compares amplicon and shotgun sequencing techniques in profiling microbial communities across diverse temperate grassland soil samples, focusing on how different databases, classifiers, and sequencing methods influence the results. Our study underscores the crucial need for a harmonized taxonomic database that could greatly enhance comparability and accuracy in the understanding of soil microbiomes.

**KEYWORDS** shotgun metagenomics, amplicon sequencing, soil microbiome, taxonomy

Assessment of the overall microbial community composition and diversity in environmental samples is achieved primarily through two strategies: (i) targeted sequencing of the 16S rRNA regions for archaea and bacteria and the ITS region for fungi, referred to as amplicon analysis, and (ii) shotgun sequencing (metagenomics), whereby all genomic DNA in the collected sample is sequenced. Currently, the most prevalent approach in environmental studies is to sequence specific hypervariable regions of the

Address correspondence to Fiona Brennan, Fiona.Brennan@teagasc.ie.

The authors declare no conflict of interest.

See the funding table on p. 14.

16S rRNA gene using short-read length platforms, typically Illumina. Both amplicon and shotgun sequencing approaches have their own sets of merits and limitations. A significant limitation of amplicon sequencing is the potential distortion of microbial profiles due to primer bias (1, 2), intragenomic variation (3), and variability in 16S rRNA gene copy numbers (4, 5) across different taxa. This variation can skew the perceived abundance of microbial taxa, complicating accurate diversity assessments. Shotgun metagenomics, while offering broader insights, is expensive compared to amplicon sequencing, computationally intensive, and prone to false identification of species by virtue of the bioinformatic tools used (6–9). Both methods share challenges related to DNA extraction variability (10) and heavy reliance on underlying databases for accurate data interpretation. The advantages of the shotgun metagenomic approach include species- or strain-level identification, functional gene profiling, and discovery of novel genomes through metagenome-assembled genomes (MAGs). MAGs are essential for exploring the largely unculturable soil microbial diversity (11). On the other hand, amplicon sequencing remains cost-effective, with simplified data analysis pipelines compared to metagenomics.

As the cost of sequencing declines, shotgun metagenomics is becoming more popular for microbiome studies. In low-diversity environments, shotgun studies report taxonomic details similar to amplicon sequencing (12, 13), making either method suitable for taxonomic profiling. However, given the high microbial diversity inherent in soils, amplicon-based assessments continue to predominate for soil studies, and additional assessments are needed for metagenomics to be more widely adopted in these environments. Previous comparisons of shotgun and amplicon sequencing in complex samples have faced limitations, such as using a suboptimal classifier for soil metagenomic analysis, neglecting potential false positives, and disregarding differences in underlying taxonomic assignment between classifiers, which can lead to biased results. The efficacy of comparing amplicon to shotgun metagenomics hinges on both the complexity of the sample and the choice of the taxonomic classifier. For example, while marker gene classifiers like MetaPhlAn are sufficient for analyzing shotgun sequencing data from less diverse microbiomes like human and chicken gut microbiomes (12, 14, 15), they struggle to capture microbial diversity in soil, lake water, or anaerobic digestion samples (1, 7, 16). This is potentially due to the higher microbial diversity prevailing in these environments together with corresponding relatively limited metagenomic efforts (17). In contrast, studies using the Kraken2 classifier avoid these discrepancies (18–20). However, even with the right classifier, it remains essential to optimize abundance thresholds to remove high false positives generated (7, 21). Indeed, comparing methods without accounting for false positives or updates to taxonomy can misleadingly indicate greater microbial richness in metagenomic data.

It is worth noting that in recent years, metagenomics has started to gain momentum in soil microbiome studies (19, 20, 22). However, few robust investigations into how amplicon and shotgun sequencing are comparable in soil systems are observed. Our research addresses this gap by evaluating shotgun and amplicon sequencing methodologies across 131 diverse soil samples collected across Ireland, with shotgun sequencing yielding an average of 20 million reads per sample. For taxonomic classification of shotgun metagenomic reads, we utilized previously validated GTDB-TK enhanced fungal database with Kraken2 and Bracken classifier, with an optimized threshold of 0.001 (7). For amplicon sequencing, we compared the SILVA and Greengenes2 databases, which showed similar prokaryotic profiles at the phylum level. We chose SILVA for its proven efficacy in classifying environmental samples, aligning our methods with standard practices in the field. For fungal identification, the UNITE database was utilized for ITS analysis. The methodological rigor employed allowed us to effectively pinpoint the origins of variations at the phylum level due to the choice of reference taxonomies, databases, classifiers, or sequencing techniques.

In this study, we focused on the following: (i) exploring the parallels and distinctions between the microbial profiles generated using the two methodologies; (ii)

examining underlying factors for any observed variances in taxonomic profiles generated; (iii) assessing the implications of differential microbial profiling for identification of relationships with microbial function.

## MATERIALS AND METHODS

### Sampling collection and DNA extraction

A total of 136 soil samples were collected across 32 managed grassland sites across Ireland (selected to incorporate a wide physicochemical and geoclimatic gradient) between 18 November 2019 and 3 April 2020 from mineral soils, as reported in Deveautour et al. (23). Further metadata on the different soil samples can be found in Table S1a (https://doi.org/10.6084/m9.figshare.27613470.v6). DNA was extracted from 0.25 g soil using the DNeasy PowerSoil Pro Kit (Qiagen, Hilden Germany) following the manufacturer's instructions. The yield and quality of DNA extracts were determined using the Qubit dsDNA BR Assay kit with an Invitrogen Qubit 3.0 fluorometer (Thermo Fisher Scientific, USA), a Nanodrop 2000 spectrophotometer (Thermo Fisher Scientific, USA), and by running samples on 1% agarose gel. DNA extracts were stored following extraction at −80°C. Originally analyzed for amplicon 16S rRNA and ITS sequencing in 2020 (23), aliquots from the same DNA extract were used for shotgun sequencing in this study.

### Amplicon library preparation and taxonomy analysis

Details on the preparation of amplicon sequencing libraries can be found in (23). Briefly, the sequencing library was prepared for the archaeal and bacterial V4 region of 16S rRNA targeted by primers 515F and 926R (24), as well as for the fungal ITS2 region targeted by primers 4R and 86F (25). First, PCR was performed with the relevant community-specific primers, in a 25 µL mix containing 2× KAPA HiFi HotStart ReadyMix (Kapa Biosystems Inc, USA), 1 µM of the primers, and 2.5 µL of DNA. Then, a second PCR was conducted with the Nextera XT index kit (Illumina, San Diego, CA) as per the manufacturer's instructions. After amplification, samples were purified using magnetic beads, before concentrated samples were pooled in equimolar ratios, diluted to 4 nM and sequenced with a 2 × 250 run on an Illumina MiSeq platform at Teagasc Next-Generation DNA sequencing facility, Ireland. The raw DNA sequencing data were processed following a similar approach to our previous work (23), with modifications to the reference databases used for taxonomic assignment. We utilized DADA2 (26) within the R environment (R version 4.0) to process both 16S and ITS sequences. For 16S rRNA sequences, the reads containing ambiguous bases were removed, and based on quality scores, forward and reverse reads were trimmed to 200 bp and 270 bp, respectively. The "dada()" function was employed for denoising, and reads were subsequently merged with a minimum overlap of 20 bp. The merged reads with lengths between 368 and 379 bp were further filtered to remove chimeras using the "removeBimera-Denovo()" function. The obtained amplicon sequence variants (ASVs) were imported into QIIME2 (version 2023.2) for taxonomic classification. We initially utilized reference databases Greengenes2 (27) and SILVA v.138.1 99% OTUs (23) for this purpose. The classifiers were NB (naive Bayes) trained by downloading the reference sequences and taxonomic data, followed by extraction of V4-specific reads. To choose the appropriate 16S rRNA classification database, we compared SILVA and Greengenes2 microbial profiles, which showed similar profiles at the phylum level (Procrustes analysis; sum of squares = 0.14, Procrustes correlation coefficient = 0.92, $P$-value < 0.001; Fig. S1 at https://doi.org/10.6084/m9.figshare.27613470.v6). Given the equivalence of both methods on samples in this study and the documented proficiency of SILVA in environmental sample classification (1, 28, 29), we opted for SILVA for the current study. For ITS sequences, no fixed length trimming was applied before denoising to accommodate the natural biological variation in ITS region length. After denoising, we examined the distribution of

merged read lengths and retained sequences between 230 and 413 bp, which correspond to the expected size range for fungal ITS2 amplicons. This post-denoising length filtering step was performed to remove potential non-target or poor-quality sequences and improve the accuracy of downstream taxonomic classification. For the analysis of fungal communities in our samples, we utilized the UNITE database (version 2023) to train a custom naïve Bayes classifier specifically for ITS region sequences, employing QIIME 2 (version 2023.2 [30]).

## Shotgun library preparation and taxonomic analysis

The Nextera XT DNA library preparation kit (Illumina, Ireland) was used to fragment DNA and add Illumina adapters. One ng DNA per sample was used to prepare the libraries following the manufacturer's instructions, which involves 12 cycles of PCR. The libraries were cleaned using Ampure XP beads (Beckman Coulter, USA), and 1 µL of each undiluted library was run on the Agilent technology bioanalyzer (Agilent Technologies, USA) using a high-sensitivity DNA chip to ensure libraries showed a broad size distribution of ~250 to 1,000 bp. The molarity of each library was then calculated based on the size of the bioanalyzer trace. The libraries were pooled to an equimolar concentration. A total of seven sequencing runs with a maximum of 21 libraries per run were sequenced. Pooled libraries of all DNA extracts were subjected to paired-end sequencing for 300 cycles with a read length of $2 \times 150$ bp across four lanes of Illumina NextSeq 2000 at Teagasc Next-Generation DNA sequencing facility, Ireland. The sequence data were analyzed using a High-Performance Computing cluster. For pre-processing, Trimgalore v0.6.1 (31) was used with parameters --paired, --fastqc, --quality 33, --threads 8. Three samples (PA17, BL5, AH2) were discarded due to their low read counts (Table S1b at https://doi.org/10.6084/m9.figshare.27613470.v6). Samples CR19 and KI12 were also excluded from further analysis due to errors during sequencing.

To analyze taxonomy from metagenomic data, we utilized a custom GTDB-Tk (32) plus fungal database developed with Kraken2 and Bracken, based on the NCBI taxonomy framework. This approach, initially validated through in silico analysis, was further corroborated using real soil metagenomic samples in our comprehensive study (7), confirming its efficacy in classifying soil metagenomes. This Kraken2 custom database, substantial in size at 1.4 terabytes, required 340–360 GB RAM. The custom database encompassed genomes from the gtdb-tk repository for bacteria and archaea (release 214). For fungi, we enhanced the database by incorporating a total of 17,637 genomes, from the group Fungi insertae sedis (major phylum: Zoopagomycota; 215 genomes, Chytridiomycota; 120 genomes, Mucoromycota; 120 genomes, and Microsporidia; 119 genomes) and the subkingdom Dikarya (major phylum: Ascomycota; 14,683 genomes and Basidiomycota; 2,362 genomes). The selection of these genomes was informed by taxonomic insights from the previously generated ITS data (23). The classifier was run on trimmed reads to obtain their taxonomic profile using a relative abundance threshold of 0.001%.

## Statistical analysis

All statistical analyses were performed in R (version 4.3.1). We processed filtered taxonomic data and generated Bray-Curtis dissimilarity matrices. These matrices were used to perform principal coordinate analysis (PCoA) using the R package vegan. We assessed the PCoA ordination similarity of the microbial profiles generated from amplicon and shotgun sequencing, using Procrustes superimposition (protest function with 999 permutations). This approach was also employed for comparing community dissimilarity between SILVA and Greengenes2 profiles. Heatmaps were created with R package ComplexHeatmap v.2.16.0, and all other figures were generated using the R package ggplot2.

We employed differential relative abundance cutoffs for data filtration across various analytical tests to address specific hypotheses. For general phylum-level microbial profiling (Fig. 1), a threshold of 0.001% relative abundance was applied to shotgun

data, whereas amplicon data were not filtered. Beta-diversity tests were conducted with tiered cutoffs of 0.1%, 0.01%, and 0.001% to evaluate the effectiveness at differing levels of microbial dominance of both methods. For heatmap visualizations, we applied a 0.01% cutoff for bacterial and archaeal phyla in both 16S rRNA amplicon and shotgun data sets, which incorporated less abundant microbes. For fungal ITS data, a 0.5% cutoff was chosen to capture all major fungal taxa effectively. For Spearman correlation assessments, no cutoffs were imposed, as the analysis was confined to phyla with high relative abundance levels.

To assess the consistency of phylum-level abundance estimates between sequencing methods, we first performed Spearman rank correlation analysis between the relative abundances of dominant phyla detected by amplicon and shotgun sequencing across all samples. Additionally, to investigate correlations between dominant phyla between the two methods, amplicon and shotgun, and phylum-level taxonomic composition and key nitrogen cycle functions (potential nitrification, potential denitrification, and the $N_2O:N_2O+N_2$ ratio), we used Spearman rank correlation. We adjusted $P$-values with Bonferroni adjustment to link these functional metrics with the relative abundance of bacterial phyla using the R package vegan (33). To statistically evaluate whether there was a significant difference between the Spearman correlation coefficients generated for the same phyla and same functions but using the two different sequencing methods, we employed Williams' $t$ test (1959) using the "cocor" package (34).

## RESULTS

### Comparative microbial profiling: shotgun and amplicon sequencing

#### *Shotgun metagenomic insights*

After quality filtering, an average of 20,419,914 sequences remained from shotgun data. Taxonomic analysis, conducted using a custom database with Kraken2 and Bracken, revealed a varied microbial composition across all soil samples (Fig. 1a). On average, 38% of the reads in each sample were classified, ranging from 17% to 50% across samples.

We identified 175 unique phyla (128, Bacteria; 2, Archaea; 7, Fungi) within our data set of 131 samples by shotgun analysis. A total of 124 phyla (approx. 70%) were classified under the "Candidatus" category, although the cumulative relative abundance of these phyla was only 2% ± 0.9% per sample. To streamline our data visualization, we consolidated these phyla into "Candidatus Bacteria" (1.9% ± 0.9%) and "Candidatus Archaea" (0.06% ± 0.05%), as illustrated in Fig. 1a. Among the bacterial "Candidatus" phylum members, "Candidatus Calescamantes" (1.41% ± 0.6%) and "Candidatus Muirbacteria" (1.55% ± 1.4%) were the most abundant, reported across 13 and 3 samples, respectively. The most prevalent was "Candidatus Rokubacteria" (0.61% ± 0.4%), detected across all 13 samples.

The most abundant bacterial phyla were Pseudomonadota (34.4% ± 2.6%), Actinomycetota (26.4% ± 4.1%), followed by Verrucomicrobiota (4.5% ± 1.4%), Bacteriodota (4.5% ± 3%), and Acidobacteriota (4.1% ± 1.4). The most abundant archaea were Euryarchaeota (0.7% ± 0.2%) and Nitrososphaerota (0.3% ± 0.4%), followed by "Candidatus Thermoplasmatota" (0.02% ± 0.003%), "Candidatus Bathyarchaeota" (0.02% ± 0.03%), and Thermoproteota (0.01% ± 0.003%). Among fungi, Ascomycota (6.5% ± 3.6%), Basidiomycota (1.9% ± 0.3%), and Mucoromycota (1% ± 1%) were the most prevalent phyla. The less prevalent fungal phyla were Zoopagomycota (0.1% ± 0.03%), Chytridiomycota (0.1% ± 0.01%), and Olpidiomycota (0.002% ± 0.0004%). Fig. 1 depicts the taxonomic profiles at the phylum level across different samples identified from shotgun data using Kraken2.

#### *Amplicon sequencing insights*

A total of 57 phyla were observed with 50 bacterial and seven archaeal phyla. The dominant bacterial phylum detected were Pseudomonadota (21.8% ± 1.8%), Actinomycetota (13.3% ± 1.8%), Acidobacteriota (12.6% ± 1.9%), Verrucomicrobiota (9.4% ± 1.6%), Planctomycetota (8.1% ± 1.4%), and Bacteriodota (6.5% ± 1.3%) across all samples (Fig.

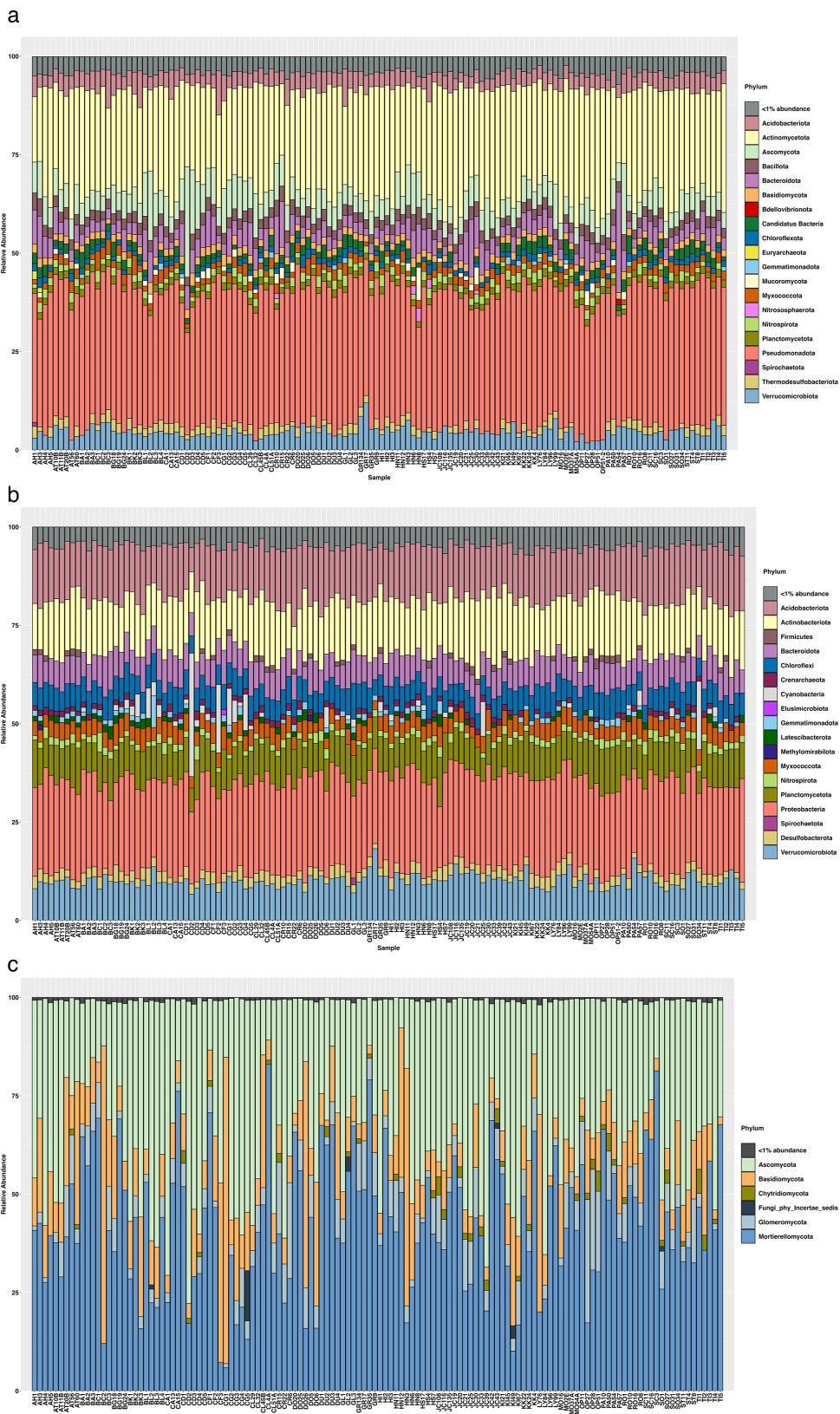

**FIG 1** Comparative analysis of soil microbial biodiversity in temperate grassland soils using both shotgun and amplicon sequencing techniques. The stacked bar plot indicates the relative abundance of identified taxa (*y*-axis) across samples (*x*-axis). (a) Shotgun-derived microbial biodiversity using the Kraken2-Bracken custom database. (b) 16S rRNA-derived microbial biodiversity using SILVA. (c) ITS-derived microbial biodiversity at the phylum level using UNITE.

1b). Dominant Archaea identified across all samples was Crenarchaeota (1.2% ± 1.2%). For the fungal community, a total of 17 phyla were observed, and the most abundant phyla were Mortierellomycota (38.2% ± 16%), Ascomycota (33.0% ± 13%), Basidiomycota (11.4% ± 12%), Glomeromycota (5.4%), and Chytridiomycota (0.88%). (Fig. 1c)

### Similarities and differences between the two methods

Our analysis revealed that the same dominant bacterial phyla were consistently identified across both amplicon and shotgun sequencing methodologies. Pseudomonadota and Actinomycetota were the most prevalent, followed by Acidobacteriota, Verrucomicrobiota, and Bacteroidota. However, we observed significant discrepancies in the identification of Archaeal phyla. Shotgun sequencing consistently identified Euryarchaeota and Nitrososphaerota, which were less detected or absent in amplicon data. Thermoproteota detection varied markedly between the two methods.

Comparison of ITS and shotgun sequencing data highlighted Ascomycota and Basidiomycota as the prominently detected phyla. However, discrepancies were evident in the representation of Mucoromycota.

## Moderate similarity in beta-diversity

To ascertain whether the relationships observed between samples based on their underlying microbial profile are conserved between methods, we calculated the beta-diversity for each of the approaches and conducted Procrustes analysis. The Procrustes correlation coefficient indicates the degree of congruence between two data sets (value of 0 indicates no relationship and value of 1 indicates complete congruence), and a low *P*-value indicates a statistically significant correlation between the ordinations. We observed a moderate level of agreement between the two approaches, with a Procrustes correlation coefficient of 0.42–0.43 (statistically significant, *P*-value < 0.001) (Table S2 at https://doi.org/10.6084/m9.figshare.27613470.v6). This suggests a moderately consistent pattern in phylum-level microbial community differentiation between the two methods. When comparing the microbial profiles from shotgun data across a variety of relative abundance cutoffs (1%, 0.1%, 0.01%, and 0.001%) to amplicon data at higher cutoffs (0.1% or 1%), the Procrustes correlation coefficient remained robust (0.42–0.43) (Fig. 2b). This suggests that shotgun sequencing maintains its effectiveness across these relative abundance thresholds. Despite the correlations observed, the analysis revealed a considerable amount of unexplained variation (high sum of squares 0.81–0.83), even in the relatively best-case scenarios where moderate agreement is observed. A dramatic shift was observed when the 16S rRNA amplicon data cutoff was reduced to below 0.01%. At lower thresholds, the correlation coefficients significantly dropped to between 0.11 and 0.14, with corresponding non-significant *P*-values (*P*-value > 0.1), implying little to no linear relationship between the data sets being compared. This highlights a stark decrease in congruence, indicating potential limitations in the ability of amplicon sequencing to capture low-abundance microbial populations.

## Taxonomic discrepancies: comparing NCBI, SILVA, and UNITE in shotgun and amplicon sequencing

Next, we compared microbial community compositions from amplicon and shotgun sequencing focusing on presence and absence patterns to highlight differences in detection between the two methods. To this end, we organized phyla into four distinct groups: (i) phyla identified by both methods and sharing the same nomenclature (19 phyla; 23%) (labeled as "Present in both" in Fig. 3); (ii) phyla detected by both methods but awaiting nomenclature updates in amplicon data sets (seven phyla; 0.1%) (labeled as "Phylum renamed" in Fig. 3), such as Proteobacteria now termed Pseudomonadota; (iii) phyla identified by both methods but subjected to taxonomic reclassification (15 phyla; 18%); for instance, Basidiobolomycota appears as a phylum in the UNITE database but is categorized under the Zoopagomycota phylum in NCBI's taxonomy (labeled as "Taxa

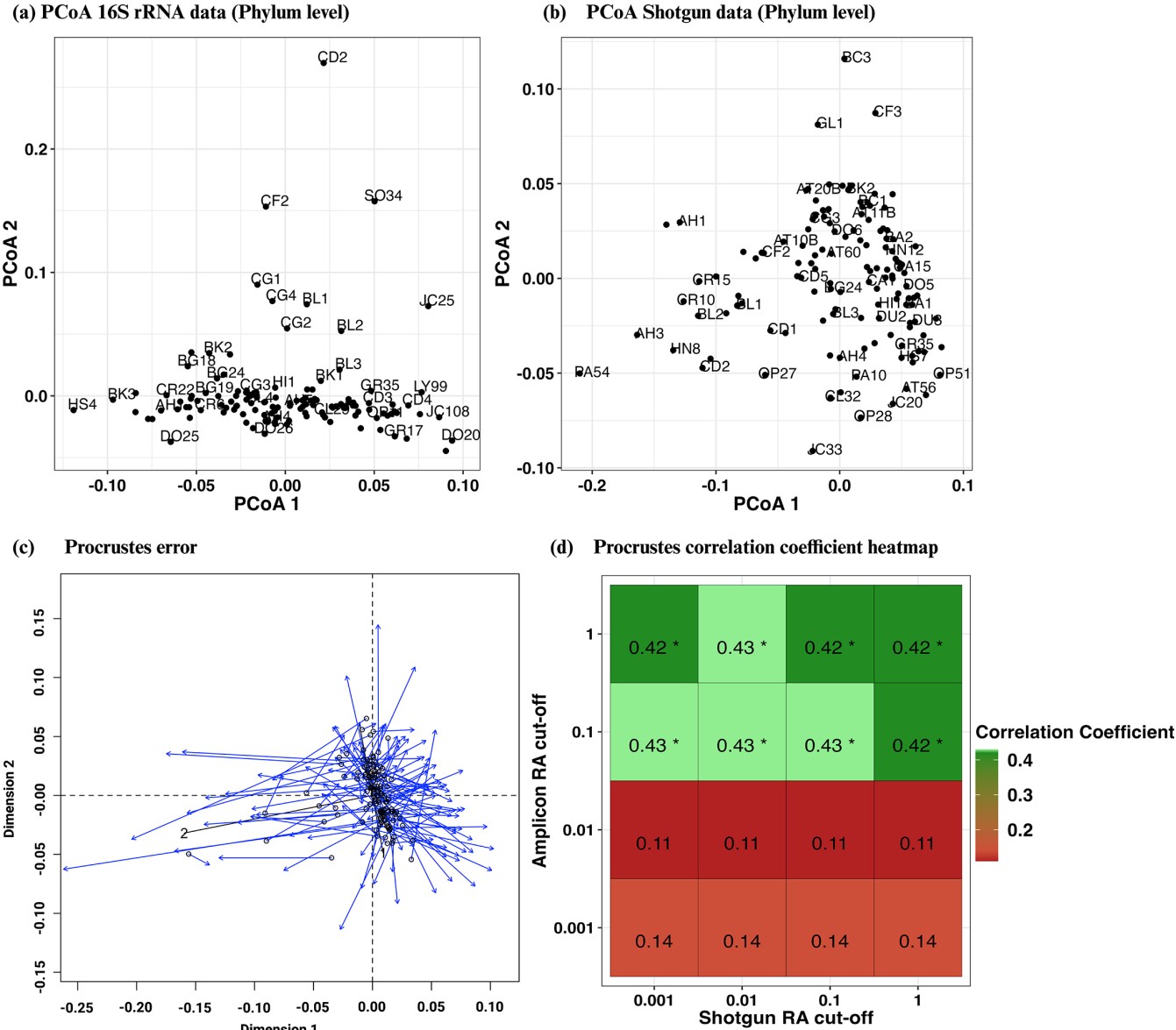

**FIG 2** Comparative principal component analysis (PCoA) plots at the phylum level for soil microbial communities and Procrustes results (a) Phylum-level beta diversity from amplicon data, presented as a PCoA plot based on Bray-Curtis dissimilarity with a 0.1% abundance threshold and (b) Corresponding PCoA plot using shotgun data. (c) Procrustes error: Ordination plot showing congruence between shotgun (data points) and amplicon data (arrow head). The shorter arrows indicate a higher degree of similarity. (d) Procrustes correlation coefficient heatmap of microbial diversity obtained from amplicon and shotgun data across different relative abundance (RA) cutoffs. Green indicates a high correlation (closer to 1), while red signifies a lower correlation (closer to 0).

reassignment" in Fig. 3). Similarly, Patescibacteria, identified in SILVA, is treated as a non-ranked group in NCBI, encompassing 57 "Candidatus" phyla; (iv) phyla detected exclusively by one method (18 by amplicon; 21% and 24 by shotgun data; 29%). These are categorized as "Absent" not detected by one method or "Present" detected by only one method in Fig. 3. Distinct microbial profiles were generated by each method, as illustrated by the clear clustering of samples according to the sequencing approach used (Fig. 3). Majority of phyla identified by one method, but not by the other, are phyla such as "Candidatus" phyla in shotgun metagenomics and NB1-j, MBNT15, RCP2-54, WS4, WS2, and WPS-2 in 16S rRNA data.

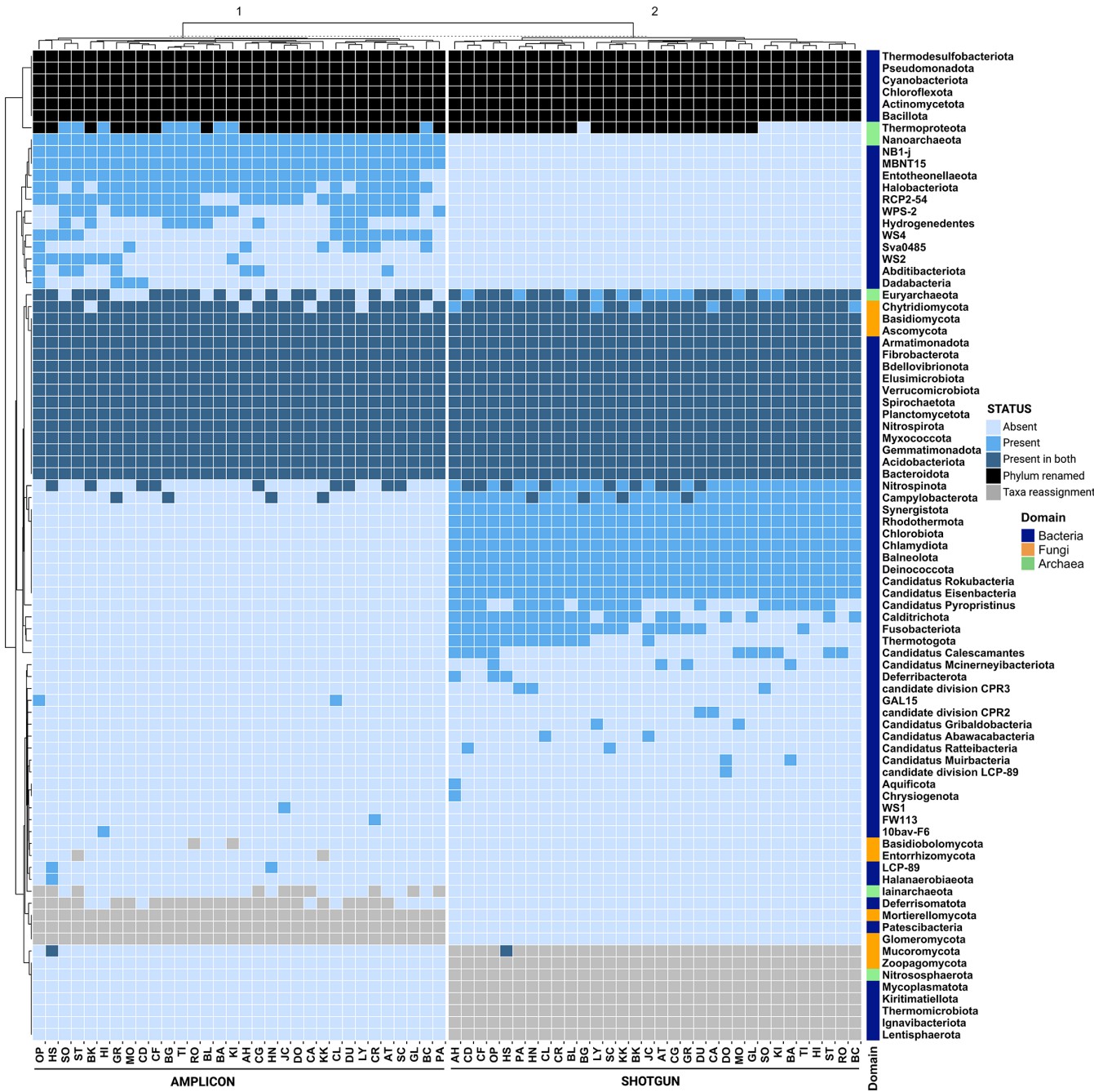

**FIG 3** Heatmap comparison of soil microbial phyla detected by amplicon and shotgun sequencing depicting the classification of soil microbial phyla, as identified by amplicon (16S rRNA and ITS data) and shotgun sequencing methods, categorized by shared and unique detections across the two sequencing methods. Clustering indicates method-specific detection patterns. The categories are as follows: Present, detected exclusively by one method; Absent, not detected by both methods or one of the methods; Present in both, detected by both methods without discrepancies; Phylum renamed, detected by both methods under different names, reflecting recent NCBI database updates; Taxa reassignment, detected by both methods, but reassigned to different taxonomic ranks (e.g., genus or class) in the NCBI database due to taxonomic updates.

## Relationships between microbial communities and nitrogen cycling functions

To assess the consistency of phylum-level abundance estimates between sequencing methods, we performed a Spearman rank correlation analysis across 17 dominant phyla including bacterial, archaeal, and fungal phyla (Table 1). Of these, eight phyla (47%)

**TABLE 1** Spearman correlation between relative abundances of dominant phyla detected by shotgun metagenomics and amplicon sequencing[a]

| Phylum | Spearman R | P-value | Significance | Kingdom |
|---|---|---|---|---|
| Verrucomicrobiota | 0.695 | 0 | <0.001 | Bacteria |
| Actinomycetota | 0.525 | 0 | <0.001 | Bacteria |
| Acidobacteriota | 0.378 | 0 | <0.001 | Bacteria |
| Spirochaetota | 0.307 | 3.00E−04 | <0.001 | Bacteria |
| Chloroflexota | 0.264 | 0.0022 | <0.001 | Bacteria |
| Basidiomycota | 0.255 | 0.003 | <0.01 | Fungi |
| Pseudomonadota | 0.228 | 0.0084 | <0.01 | Bacteria |
| Ascomycota | 0.223 | 0.0098 | <0.01 | Fungi |
| Planctomycetota | 0.122 | 0.1602 | ns | Bacteria |
| Cyanobacteriota | 0.089 | 0.3092 | ns | Bacteria |
| Thermodesulfobacteriota | 0.065 | 0.4539 | ns | Bacteria |
| Thermoproteota | 0.028 | 0.7456 | ns | Archaea |
| Bacillota | 0.026 | 0.7637 | ns | Bacteria |
| Bacteroidota | 0.009 | 0.9193 | ns | Bacteria |
| Myxococcota | −0.006 | 0.9457 | ns | Bacteria |
| Chytridiomycota | −0.014 | 0.8734 | ns | Fungi |
| Nitrospirota | −0.037 | 0.6733 | ns | Bacteria |
| Mucoromycota | −0.077 | 0.3757 | ns | Fungi |

[a] ns = non-significant; any P-value > 0.01.

showed statistically significant correlations in relative abundance between amplicon and shotgun data, including Verrucomicrobiota ($\rho = 0.695$, $P < 0.001$), Actinomycetota ($\rho = 0.525$, $P < 0.001$), Basidiomycota ($\rho = 0.255$, $P = 0.003$), and Ascomycota ($\rho = 0.223$, $P = 0.0098$). The remaining phyla displayed low correlation coefficients and nonsignificant P-values, suggesting variable agreement in relative abundance values between methods. These results indicate that while both methods identify similar patterns in phylum prevalence, the relative abundance values are not directly comparable.

Building on the above, we explored whether the ecological relationships between microbial phyla and nitrogen cycling functions (as assessed via potential nitrification, denitrification, and the $N_2O:N_2O+N_2$ ratio) are conserved across sequencing methods, despite differences in relative abundance values. The comparative use of Spearman rank correlations allowed us to systematically evaluate how shifts in microbial phyla abundance, detected through each sequencing method, correlated with key nitrogen cycle processes and to determine whether such relationships were conserved across sequencing approaches.

We focused on a range of phyla, including Acidobacteriota, Actinomycetota, Bacteriodota, Bacillota, Chloroflexota, Thermoproteota/ Crenarchaeota (Archaea), Cyanobacteriota, Thermodesulfobacteriota, Myxococcota, Nitrospirota, Planctomycetota, Pseudomonadota, Spirochaetota, and Verrucomicrobiota and fungal phylum Ascomycota, Basidiomycota, Mucoromycota, and Chytridiomycota. These phyla were chosen due to their abundance, and some had been previously well-characterized for their involvement in nitrogen cycling (35–37). Initial correlations revealed that while there were differences in relative abundances reported for these phyla using amplicon and shotgun metagenomic methods, the overall trends in the correlation were similar for all the three functional activities tested (potential nitrification rates, potential denitrification rates, and the $N_2O:N_2O+N_2$ ratio) (Fig. S3a to c at https://doi.org/10.6084/m9.figshare.27613470.v6). Notably, the phyla Actinomycetota and Cyanobacteriotota were significantly correlated with potential nitrification rates, as identified by shotgun (Actinomycetota $R = 0.39$, P-value $< 0.001$, Cyanobacteriota $R = -0.3$, P-value $< 0.01$), with Actinomycetota similarly indicated by amplicon analysis ($R = 0.3$, $P < 0.01$). In addition, Verrucomicrobiota was found to be significantly positively correlated to the ratio of $N_2O:N_2O+N_2$ ratio released, by both sequencing methods ($R = 0.32$, P-value $<$

0.01; by amplicon, $R$ = 0.31, $P$-value < 0.01; by shotgun data). In contrast, correlations for denitrification rates for phyla Actinomycetota ($R$ = 0.4; $P$-value < 0.001) and Chloroflexota ($R$ = 0.47; $P$-value < 0.001) emerged solely from shotgun sequencing data. The fungal phylum, Basidiomycota, however, was significantly negatively correlated to potential nitrification rates, and this trend was picked up by both sequencing methods ($R$ = −0.28, $P$-value < 0.01; by amplicon, $R$ = −0.34, $P$-value < 0.001; by shotgun data).

Using William's test, we also identified few notable disparities in Spearman correlation coefficients between microbial phyla and environmental variables across sequencing methodologies (Table S3a to c at https://doi.org/10.6084/m9.figshare.27613470.v6). Notably, the potential nitrification rates and the relative abundance of Bacillota showed divergent correlations: a positive correlation in amplicon data ($R$ = 0.12) versus a negative correlation in shotgun data ($R$ = −0.27). This difference was statistically significant ($P$ < 0.01), suggesting a method-dependent bias in correlating microbial abundance with nitrification rates. Other bacterial phyla such as Cyanobacteriota and Planctomycetota, and the fungal phylum Chytridiomycota, also exhibited disparities in nitrification correlations between methods, although these were less pronounced than those observed in Bacillota. Actinomycetota demonstrated a markedly stronger positive correlation with potential denitrification rates in shotgun data ($R$ = 0.4) compared to amplicon data ($R$ = 0.16), with a significant difference ($P$ < 0.01). Additionally, Acidobacteriota and Nitrospirota showed significant variations in their correlations with $N_2O:N_2O+N_2$ ratios across the two methods (amplicon $R$ = 0.021 vs shotgun $R$ = 0.24; amplicon $R$ = 0.1 vs shotgun $R$ = −0.17, respectively) with $P$-values < 0.05.

## DISCUSSION

In this study, we conducted a comprehensive comparison of microbial community profiles in diverse Irish soil samples, utilizing both shotgun and amplicon sequencing (targeting the V4 region of 16S rRNA and the ITS region). Initially, we observed significant variations in community composition and richness of microbial phyla detected by each method. Specifically, shotgun sequencing identified a substantially higher number of unique phyla; 175 in total (128 bacteria, two archaea, and seven fungi), compared to the 64 phyla (50 bacteria, seven archaea, and 17 fungi) detected using amplicon sequencing, although the relative abundance of these differentially occurring phyla was generally low. However, in a detailed analysis with taxonomic discrepancies between both methods appropriately managed, we found that differences in microbial profiling were less substantial than previously believed as most of the variations were attributed to the choice of reference taxonomies. All comparisons were made at the phylum level to delineate and identify the sources of differences between the two methods investigated. The comparisons at the phylum level revealed moderate similarities between the two methods investigated (Fig. 2; Table 1). We also assessed the impact of taxonomic classification differences on the analysis of soil samples, and despite the discrepancies, our findings indicate that both sequencing methods preserved moderate consistency in diversity metrics. Importantly, this consistency extended to the relationships between samples and with environmental variables, suggesting that fundamental ecological insights remained robust across methodologies.

Our study confirms that both sequencing methods successfully recovered the most abundant phyla, corroborating findings from previous studies (19, 20, 38). For fungi, although the phyla Ascomycota and Basidiomycota showed similar prevalence trends in both methods (39), the phylum Mucoromycota was underrepresented in the shotgun sequencing data. This discrepancy likely stems from the limited representation of Mucoromycota genomes in the NCBI database. While our database included a comprehensive collection of genomes for Ascomycota (14,683) and Basidiomycota (2,362), it contains only 418 for Mucoromycota, potentially leading to less effective recovery in shotgun analyses. This issue highlights a significant challenge in accurately detecting Mucoromycota, reflecting the current limitations of genomic databases. Interestingly,

despite a similar underrepresentation of Mucoromycota in ITS databases (40), the targeted approach of ITS sequencing likely mitigates this issue in amplicon studies.

In soil microbial ecology, a fundamental challenge is identifying divergence in microbial communities between samples. This differentiation is essential for effectively assessing how various environmental factors influence these microbial ecosystems. Our study demonstrates that while 16S rRNA and shotgun sequencing methods are moderately congruent in identifying microbial communities at higher-abundance thresholds, discrepancies arise at lower ones, which could potentially affect the interpretations of data concerning less abundant taxa. We describe the relationship between these methods as having moderate similarity, as quantified by Procrustes correlation coefficients ranging from 0.42 to 0.43. Unlike the previous study on on-site wastewater systems that showed high congruence between the methods (18), our findings indicate comparatively less congruity. Additionally, our Spearman correlation analysis revealed that although nearly half of the dominant phyla exhibited statistically significant correlations between methods, many others did not. This supports our interpretation that the community structure is moderately similar, yet the relative abundance values of individual phyla differ. As such, we propose that microbial associations with nitrogen functions should be viewed as qualitatively but not quantitatively consistent when comparing amplicon and shotgun approaches. We report that shotgun sequencing maintains consistent community interpretations across varying abundance thresholds, whereas 16S rRNA amplicon sequencing diverges considerably as these thresholds lower. This highlights the potential limitation of the method in ecological studies when analyzing less abundant taxa. We also demonstrate a similar concordance between dominant microbial phyla and their relationship to key nitrogen cycle functions. Both sequencing methods showed consistent results across most phyla with potential nitrification, denitrification, and $N_2O:N_2O+N_2$ emissions (14 out of 18, 15 out of 18, and 16 out of 18, respectively). Significant correlations were observed for microbial taxa known to be associated with specific functions, e.g., Actinomycetota (35, 36), and potential nitrification rates. While both sequencing approaches are effective at broader taxonomic scales, deeper insights might be gleaned using methods focusing on functional profiling. Additionally, although our shotgun genomics approach achieved sufficient sequencing depth (~20 million reads) for taxonomic analysis, it remains uncertain whether this depth is adequate for obtaining long contigs necessary for a comprehensive functional profile, a question that was not directly addressed in this study.

Despite observing a general consistency across various metrics, our analysis highlighted discrepancies, particularly in relation to how taxonomy databases influence our interpretations. Comparing shotgun metagenomics (NCBI taxonomy) and amplicon data sets (SILVA taxonomy), we noted an initial overlap of ~22%, which increased to ~50% when adjusted for taxonomy differences. This recalibration provides a more accurate comparison of the phyla detected by the two methods, indicating a closer methodological alignment than initially observed. This underscores the need for caution when comparing the two methods as attributing superiority based on taxonomic richness could be misleading due to the comprehensiveness of the respective databases. While a potential solution could be to harmonize the reference database used across both data sets, the only database currently available for both, Greengenes2, relies on a marker gene approach, which might be inadequate for comprehensive soil shotgun metagenomic analysis. Consequently, these taxonomy differences significantly limited our ability to compare microbial communities at finer taxonomic resolutions, where discrepancies are more pronounced by the increased complexity and scale.

Previous research has suggested that as much as 73% of taxa at the phylum level may be unique to one database, SILVA, RDP, Greengenes2, or NCBI, with the highest concordance (60%) observed between SILVA and NCBI (41). Our analysis supports this notable overlap between these databases. Update frequencies of these databases differ markedly; as of May 2024, SILVA released version 138 in 2019, followed by an update in

2020. In contrast, NCBI has regular updates, ensuring alignment with major taxonomic revisions. As of January 2023, NCBI incorporated extensive modifications to 42 phylum names following the latest guidelines of the International Code of Nomenclature for Prokaryotes (ICNP) (42, 43). These substantial changes in taxonomy represent a key challenge in achieving consistent comparative results across studies using different databases. This issue is further complicated by the regular reclassification of "Candidatus" phyla in the NCBI database, often represented through MAGs. The difficulty in obtaining 16S rRNA gene copies from these incomplete genomes is well-documented (44–46), suggesting that amplicon data sets might lag behind for these taxa. Additionally, we hypothesize that some observed discrepancies in microbial profiles, notably the enhanced detection of Pseudomonadota and Actinomycetota, may not solely arise from the taxonomic framework. These patterns could be influenced by the classifier, the sequencing technique, or a combination of both, as similarly reported in previous soil sequencing studies using Kraken2 (19, 20, 22). By systematically comparing amplicon and shotgun sequencing approaches at the phylum level, our study quantifies discrepancies and highlights the impact of taxonomy and tools like Kraken2 on classifying soil microbial communities.

While our study provides important insights into the comparability of amplicon and shotgun sequencing, several limitations should be acknowledged. First, amplicon sequencing relies on domain-specific primers (specifically the V3 region of 16S and the ITS region) that may not equally amplify all microbial taxa. As such, underrepresentation of certain groups, particularly those with mismatches to primer regions, is likely attributable to primer bias, a known limitation in marker gene studies (1, 2). Second, our correlation and functional analyses were intentionally limited to a subset of dominant phyla that were consistently detected across both methods and reference databases. This decision was made to ensure interpretability and reduce the confounding effects of database-inconsistent or low-abundance taxa, which are often inconsistently classified, especially for "Candidatus" lineages or taxa primarily represented by MAGs. Including all phyla would have introduced noise and obscured meaningful ecological trends. By focusing on reliably detected phyla, we aimed to ground our comparisons in robust signals rather than taxonomic artifacts.

Our study revealed that amplicon and shotgun sequencing can provide comparable outcomes when taxonomic discrepancies are considered. To effectively compare microbial profiles from metagenomic and amplicon data, precise tools such as optimally calibrated classifiers for soil microbiomes, suitable relative abundance thresholds, and curated databases are essential (7). Unlike amplicon data analysis, developing and utilizing such a custom database, particularly a k-mer-based approach like Kraken2, demands substantial computational resources, which may complicate comparative studies. This contrasts with the findings of various methods employed for amplicon data analysis, including both alignment-based approaches and probabilistic classifiers (47), highlighting the need for method-specific analytical tools for accurate comparisons. Employing this approach allowed us to identify potential biases associated with sequencing method, reference taxonomies, and classifiers, highlighting the importance of our methodology in future research for accurate comparative studies. Our findings guide us toward refining these analytical tools for soil-specific microbial data and underscore the importance of employing a unified taxonomic database to ensure effective comparison of microbial taxa from both data sets. Supported by initiatives like SeqCode (48) and GTDB-Tk (32), harmonizing taxonomy files across major databases like SILVA and NCBI in the future is crucial for enhancing the interoperability and comparability of microbial studies (49). Additionally, future studies should also determine the sequencing depth necessary not only for accurate taxonomic analysis but also for effective functional profiling in shotgun sequencing from soil samples. Our study reports amplicon sequencing as a viable and cost-effective option when databases are frequently updated, while shotgun sequencing offers access to the latest databases and provides species- or strain-level classification. The choice between these two methods

should consider the project requirements, database comprehensiveness, access to high computing facilities, and budget constraints.

## Conclusion

Despite thorough analysis, the limitations of our comparative approach, along with the absence of a clear baseline for validation in real-world data (as opposed to *in silico* studies), make it difficult to determine whether shotgun or amplicon sequencing is superior for profiling microbial communities. In conclusion, it remains clear that neither shotgun nor amplicon sequencing can be considered a definitive gold standard for microbial community profiling. Each method presents unique advantages and disadvantages. For instance, while shotgun sequencing can identify a wider range of microorganisms, it may not consistently provide the depth needed for detecting less abundant taxa as effectively as amplicon sequencing. On the other hand, amplicon sequencing, despite its high sensitivity for certain taxa, depends heavily on the choice of primers and the regions targeted, which can bias the results toward certain microbial groups. These method-dependent biases highlight the complexity of accurately representing microbial communities and underscore the necessity for developing a more standardized approach to microbial profiling. Future research should aim at enhancing methodological congruence and addressing the disparities in database completeness and update frequencies, which are critical for advancing our understanding of microbial ecosystems.

## ACKNOWLEDGMENTS

We thank Dr. Fiona Crispie for amplicon and metagenomic sequencing. We thank Dr. Paul Cormican for all the technical support provided during the course of study. We further thank Paula Rojas-Pinzon, James Rambaud, and the technical teams in Teagasc Johnstown Castle who contributed to the development of the datasets used in this study.

The shotgun metagenomic work was supported by the Teagasc Walsh Scholarship Programme (Ref: 2020019) and Science Federation Ireland (SFI) together with the Irish Department of Agriculture, Food and Marine (DAFM) under grant number SFI/16/RC/3835 (VistaMilk). The initial amplicon study and sample collection was financially supported under the National Development Plan, through the Research Stimulus Fund, administered by the Irish Department of Agriculture, Food and the Marine (Grant number 15S655: MINE project).

## AUTHOR AFFILIATIONS

[1]Teagasc, Moorepark Food Research Centre, Fermoy, County Cork, Ireland
[2]Functional Environmental Microbiology, University of Galway, Galway, County Galway, Ireland
[3]VistaMilk Science Foundation Ireland (SFI) Research Centre, Cork, Ireland
[4]Soils, Environment and Landuse Department, Teagasc, Wexford, Ireland
[5]AGHYLE Research Unit, Institut Polytechnique UniLaSalle, Mont-Saint-Aignan, France

## AUTHOR ORCIDs

Niranjana Rose Edwin http://orcid.org/0000-0002-8433-7676
Fiona Brennan http://orcid.org/0000-0002-2949-6180

## FUNDING

| Funder | Grant(s) | Author(s) |
| --- | --- | --- |
| Teagasc – the Agriculture and Food Development Authority | 2020019 | Fiona Brennan |
| | | Florence Abram |
| | | Orla O'Sullivan |

| Funder | Grant(s) | Author(s) |
|---|---|---|
| Department of Agriculture, Food and the Marine, Ireland | SFI/16/RC/3835 | Fiona Brennan |
| | | Florence Abram |
| | | Orla O'Sullivan |
| Department of Agriculture, Food and the Marine, Ireland | 15S655 | Fiona Brennan |

## AUTHOR CONTRIBUTIONS

Niranjana Rose Edwin, Conceptualization, Data curation, Formal analysis, Investigation, Visualization, Writing – original draft | Aoife Duff, Methodology, Writing – review and editing | Coline Deveautour, Formal analysis, Methodology, resources, Writing – review and editing | Fiona Brennan, Funding acquisition, Project administration, Supervision, Writing – review and editing | Florence Abram, Funding acquisition, Project administration, Supervision, Writing – review and editing | Orla O'Sullivan, Funding acquisition, Project administration, Supervision, Writing – review and editing

## DATA AVAILABILITY

The codes utilized for preparation of reference databases and training of classifiers for SILVA, Greengenes2 (amplicon and shotgun), and ITS can be found at https://forum.qiime2.org/t/processing-filtering-and-evaluating-the-silva-database-and-other-reference-sequence-data-with-rescript/15494, https://forum.qiime2.org/t/introducing-greengenes2-2022-10/25291, and https://john-quensen.com/tutorials/training-the-qiime2-classifier-with-unite-its-reference-sequences/. The 16S and ITS DNA sequencing data are available on the NCBI database under the accession number BioProject PRJNA788893, and the shotgun metagenomic data are available in ENA under Project Accession PRJEB75964. The list of genomes used to construct the custom Kraken database for archaea and bacteria can be accessed at https://data.gtdb.ecogenomic.org/releases/release214/214.1/. Additionally, the accession IDs for the fungal genomes incorporated into the database are detailed in Table S7 at https://doi.org/10.6084/m9.figshare.27613470.v6. All associated supplemental tables and figures are available at https://doi.org/10.6084/m9.figshare.27613470.v6.

## ADDITIONAL FILES

The following material is available online.

### Open Peer Review

PEER REVIEW HISTORY (review-history.pdf). An accounting of the reviewer comments and feedback.

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
