## [Reviewer comments · mSystems]

Consistent microbial insights across sequencing methods in soil studies: The role of reference taxonomies

Niranjana Edwin, Aoife M. Duff, Coline Deveautour, Fiona Brennan, Florence Abram, and Orla O'Sullivan

Corresponding Author(s): Fiona Brennan, TEAGASC

Review Timeline:

Submission Date:	August 6, 2024
Editorial Decision:	October 22, 2024
Revision Received:	November 25, 2024
Editorial Decision:	March 17, 2025
Revision Received:	May 11, 2025
Accepted:	May 12, 2025

Editor: Rachel Poretsky

Reviewer(s): The reviewers have opted to remain anonymous.

Transaction Report:

DOI: <https://doi.org/10.1128/msystems.01059-24>

Re: mSystems01059-24 (Consistent microbial insights across sequencing methods in soil studies: The role of reference taxonomies)

Dear Dr. Fiona Brennan:

This paper should be a useful contribution to the field. There are some minor edits and suggested analyses that should be done prior to acceptance.

Revision Guidelines

Sincerely,
Rachel Poretsky
Editor
mSystems

Reviewer #1 (Comments for the Author):

Overall this is a comprehensive study of different sequencing-based approaches to understanding the highly complex soil microbiome.

The data and statistical approaches are appropriate and explained well. There are only a few small things that I think should be considered.

1) I think there should be a clear acknowledgement in the conclusions that one of the possible problems is the lack of a gold-standard to compare the results to. While one method may appear superior than another in a comparison, such as that carried out, it is likely that there are pros and cons associated with each that a correlative analysis between two incomplete overviews will not identify.

2) What is the possible impact of difference in 16S rRNA gene copy number on the results? There was no discussion of this important topic, which should be acknowledged.

3) It might have made sense to carry out a PiCRUST analysis of the functions predicted from the 16S rRNA profiles, to compare to the metagenomic functions as this is what is commonly carried out with 16S rRNA data. This was not even mentioned in the paper, which seems like an oversight for a paper that is trying to provide guidance to the field.

4) Please provide in the supplementary the list and IDs of all extra genomic data added to the reference databases as without this the analysis is not replicable, even better would be to host the databases used somewhere. This could be provided as a link to an online repository for the paper on something like figshare or the Open Science Framework.

Table of Contents

Reviewer 1 (Comments for author):	1
Reviewer 1: comment 1:	1
Reviewer 1: comment 2:	2
Reviewer 1: comment 3:	2
Reviewer 1: comment 4:	2

Reviewer 1 (Comments for author):

Overall this is a comprehensive study of different sequencing-based approaches to understanding the highly complex soil microbiome.

The data and statistical approaches are appropriate and explained well. There are only a few small things that I think should be considered.

We thank the reviewer for taking time to review our manuscript and their kind words. The specific queries flagged by the reviewer have been explained thoroughly below and have been addressed within the manuscript.

Reviewer 1: comment 1:

1) I think there should be a clear acknowledgement in the conclusions that one of the possible problems is the lack of a gold-standard to compare the results to. While one method may appear superior than another in a comparison, such as that carried out, it is likely that there are pros and cons associated with each that a correlative analysis between two incomplete overviews will not identify.

We thank the reviewer for the insightful comment regarding the necessity of acknowledging the absence of a definitive gold standard for comparing microbial community profiling methods.

In response to the suggestion, we have revised our concluding paragraph to emphasize that neither shotgun nor amplicon sequencing can be considered as the definitive gold standard for microbial community profiling in lines 520-535

Lines 520-535:

CONCLUSION

Despite thorough analysis, the limitations of our comparative approach, along with the absence of a clear baseline for validation in real-world data (as opposed to *in silico* studies), make it difficult to determine whether shotgun or amplicon sequencing is superior for profiling microbial communities. In conclusion, it remains clear that neither shotgun nor amplicon sequencing can be considered a definitive gold standard for microbial community profiling. Each method presents unique advantages and disadvantages. For instance, while shotgun sequencing can identify a wider range of microorganisms, it may not consistently provide the depth needed for detecting less abundant taxa as effectively as amplicon sequencing. On the other hand, amplicon sequencing, despite its high sensitivity for certain taxa, depends heavily on the choice of primers and the regions targeted, which can bias the results towards certain microbial groups. These method-dependent biases highlight the complexity of accurately representing microbial communities and underscore the necessity for developing a more standardized approach to microbial profiling. Future research should aim at enhancing methodological congruence and addressing the disparities in database completeness and update frequencies, which are critical for advancing our understanding of microbial ecosystems.

Reviewer 1: comment 2:

2) What is the possible impact of difference in 16S rRNA gene copy number on the results? There was no discussion of this important topic, which should be acknowledged.

We thank the reviewer for making this point. We have addressed this consideration in the introduction (Lines 75-78) where we discuss the limitations of amplicon sequencing and in the discussion (Lines 422- 429).

Lines 75 - 78: A significant limitation of amplicon sequencing is the potential distortion of microbial profiles due to primer bias [1, 2], intragenomic variation [3] and variability in 16S rRNA gene copy numbers [4, 5] across different taxa. This variation can skew the perceived abundance of microbial taxa, complicating diversity assessments.

Reviewer 1: comment 3:

3) It might have made sense to carry out a PiCRUST analysis of the functions predicted from the 16S rRNA profiles, to compare to the metagenomic functions as this is what is commonly carried out with 16S rRNA data. This was not even mentioned in the paper, which seems like an oversight for a paper that is trying to provide guidance to the field.

We thank the reviewer for the suggestion regarding the potential inclusion of PICRUSt analysis to predict functional profiles from 16S rRNA data. We acknowledge the relevance of the point, raised particularly given the common use of PICRUSt in complementing 16S rRNA studies by inferring functional capabilities. However, after careful consideration, we opted not to include PICRUSt analysis in our study for several specific reasons aligned with our research objectives and the broader scientific context:

Focus on Taxonomic and Methodological Comparison: The core aim of our research is to address the ongoing challenges related to taxonomic database discrepancies and to evaluate the comparability of shotgun and amplicon sequencing methods. Integrating a PICRUSt analysis, which primarily serves to infer functional potential from taxonomic data, could divert focus from these taxonomic and methodological issues. Our study seeks to provide clarity on the direct observational capabilities of these sequencing techniques without the added layer of inferred functional analysis.

Relevance: Prior studies, including those by Kong et al. (2023)[6] and Sun et al. (2020)[7], have highlighted critical limitations in the accuracy of PICRUSt predictions, particularly when applied outside of human-associated microbiomes. Wong et al. (2016) [8] highlighted that PICRUSt's predictions might be too generalized to distinguish between different metabolic pathways, necessitating the higher resolution offered by shotgun metagenomics for detailed functional profiling.

These studies have shown that PICRUSt may inaccurately assign specific pathways to bacteria, potentially misrepresenting their functional roles in complex ecosystems. Given that extensive research has already been conducted comparing PICRUSt with shotgun metagenomics, including findings that question the reliability of PICRUSt predictions in non-human and environmental samples, we felt that revisiting this analysis would not add novel insights to our study.

Reviewer 1: comment 4:

4) Please provide in the supplementary the list and IDs of all extra genomic data added to the reference databases as without this the analysis is not replicable, even better would be to host the databases used somewhere. This could be provided as a link to an online repository for the paper on something like figshare or the Open Science Framework.

We agree with the reviewer about the importance of transparency and replicability in research. However, the large size of the database (212 GB in its most compressed form) presents significant financial challenges for hosting it on a public platform. We are actively exploring options to secure funding to make the database publicly accessible, but this process is still ongoing.

In the interim, we have ensured that all relevant genomic metadata, including accession IDs, are accessible. Detailed information on how to access these genomes is provided in the “Data Availability” section of our manuscript.

Specifics on Data Listing (Lines 596-601): The list of genomes used to construct the custom Kraken database for archaea and bacteria can be accessed at <https://data.gtdb.ecogenomic.org/releases/release214/214.1/>. Additionally, the accession IDs for the fungal genomes incorporated into the database are detailed in Supplementary Table 8. All manuscript associated supplementary tables are available at <https://doi.org/10.6084/m9.figshare.27613470.v1>

1. Campanaro, S., et al., *Taxonomy of anaerobic digestion microbiome reveals biases associated with the applied high throughput sequencing strategies*. Scientific reports, 2018. **8**(1): p. 1926.
2. Jeong, J., et al., *The effect of taxonomic classification by full-length 16S rRNA sequencing with a synthetic long-read technology*. Scientific reports, 2021. **11**(1): p. 1-12.
3. Sun, D.-L., et al., *Intragenomic heterogeneity of 16S rRNA genes causes overestimation of prokaryotic diversity*. Applied and environmental microbiology, 2013. **79**(19): p. 5962-5969.
4. Větrovský, T. and P. Baldrian, *The variability of the 16S rRNA gene in bacterial genomes and its consequences for bacterial community analyses*. PloS one, 2013. **8**(2): p. e57923.
5. Gao, Y. and M. Wu, *Accounting for 16S rRNA copy number prediction uncertainty and its implications in bacterial diversity analyses*. ISME communications, 2023. **3**(1): p. 59.
6. Kong, Y., D. Li, and X. Meng, *Gut microbiota dysbiosis in rats with LPS-induced liver diseases affected by Aronia melanocarpa polyphenols*. Food Science and Technology, 2023. **43**.
7. Sun, S., R.B. Jones, and A.A. Fodor, *Inference-based accuracy of metagenome prediction tools varies across sample types and functional categories*. Microbiome, 2020. **8**: p. 1-9.
8. Wong, K., et al., *Rapid microbiome changes in freshly deposited cow feces under field conditions*. Frontiers in microbiology, 2016. **7**: p. 500.

Re: mSystems01059-24R1 (Consistent microbial insights across sequencing methods in soil studies: The role of reference taxonomies)

Dear Dr. Fiona Brennan:

Thank you for the privilege of reviewing your work. Apologies that it took so long to secure reviews. Below you will find my comments, instructions from the mSystems editorial office, and the reviewer comments. You will see a request for minor modifications. After these are completed, your article will be accepted.

Revision Guidelines

Sincerely,
Rachel Poretzky
Editor
mSystems

Reviewer #2 (Comments for the Author):

This study compares amplicon and shotgun metagenomic sequencing approaches to analyze microbial communities in 131 temperate grassland soil samples from Ireland. The authors assess how these methods differ in detecting and classifying microbial taxa, their ability to distinguish microbial communities across samples, and their capacity to link microbial communities to nitrogen cycle functions. The study finds that while both methods broadly identify similar microbial patterns, discrepancies arise mainly due to differences in taxonomic classification methods. Shotgun sequencing provides deeper taxonomic resolution

but requires more computational resources, whereas amplicon sequencing is a cost-effective and widely used alternative. The findings emphasize the importance of selecting an appropriate sequencing approach based on study objectives and highlight the need for harmonized taxonomic databases for improved comparability of soil microbiome studies.

This study is well-structured and provides a comprehensive comparison of sequencing methodologies used in soil microbial ecology. The large sample size strengthens the reliability of the findings, and the methodological rigor, particularly the use of multiple reference taxonomies and classifiers, enhances the credibility of the results. However, several areas could be improved to enhance the clarity and impact of the manuscript.

First, the correlation between the relative abundances of phyla detected by shotgun metagenomics and those detected by amplicon sequencing is not clearly presented, making the discussion ambiguous. While the manuscript states that the results from shotgun and amplicon sequencing are "similar," this assessment primarily focuses on the presence or absence of phyla rather than evaluating how well the relative abundances of each phylum match between the two methods. This is a fundamental issue in interpreting the study's findings. If the relative abundances between shotgun and amplicon sequencing were consistent, then the correlations between microbial relative abundances and nitrogen cycle-related environmental factors, as discussed in the latter part of the manuscript (L375-386), should also be consistent across both methods. Conversely, if the relative abundances differ significantly between the two approaches, the premise for discussing correlations with nitrogen cycling is undermined. Therefore, before engaging in discussions on the relationship between microbial communities and nitrogen cycling functions, the manuscript should first establish the correlation between shotgun and amplicon sequencing results. As it stands, the discussion on nitrogen cycle-related functions is likely not addressing the most critical issue.

Second, the manuscript calculates the ratio of microbial groups detected by amplicon sequencing and shotgun metagenomics based on the equation in L227-229, but it does not justify the validity of this approach. This justification is essential because amplicon sequencing targets specific domains (e.g., bacteria and archaea) using domain-specific primers, whereas shotgun metagenomics sequences all DNA present in the sample, including eukaryotic and prokaryotic DNA. As a result, a direct comparison of relative abundances between the two methods may not be appropriate without proper normalization. To ensure the robustness of the analysis, the manuscript should either (i) provide a clear rationale for why this ratio calculation is valid or (ii) discuss the potential biases introduced by this method and consider whether corrections are needed.

Furthermore, the use of Bray-Curtis dissimilarity for MDS (Multidimensional Scaling) analysis (L318-330) may also be affected by this issue. Bray-Curtis dissimilarity is typically applied to non-binary community composition data, and differences in data characteristics between shotgun and amplicon sequencing may introduce biases into the MDS results. The authors should either justify the validity of applying Bray-Curtis dissimilarity to these datasets or consider alternative normalization approaches or distance metrics to ensure that the comparison is not biased by methodological differences between the two sequencing techniques. On the other hand, analyses using rank-based correlations to assess the relationship between microbial communities and nitrogen cycling functions are less likely to be affected by the above concerns. However, to strengthen the study's reliability, it is still crucial to first clarify the extent to which shotgun and amplicon sequencing data are correlated before interpreting their functional associations.

Addressing these points would improve the clarity of the discussion and enhance the robustness of the study's conclusions.

Specific Comments

- L114: Specify whether this refers to per sample or total across all samples.
- L275-277: Remove "at" before the percentage values to improve readability.
- L176: Explain why these sequence lengths were chosen.
- L183: How many cycles of PCR were performed?
- L200: The reference information appears to be incorrect. Please correct it.
- L306: Should be Table 2.
- L345: What do categories ii and iii refer to in the status shown in Fig. 3?
- L408: In cases where the relative abundance obtained from amplicon sequencing is lower than that from shotgun sequencing, primer coverage issues are likely involved. This aspect should also be included in the discussion.
- L414: References are needed here.
- L470: What does "standardise" mean in this context? Is it referring to nomenclature?
- L513: Since this study does not examine sequencing depth, it may be omitted.
- L519: Amplicon sequencing can also provide species-level taxonomy.

Reviewer #4 (Comments for the Author):

The revised manuscript is well-written and provides clear insights into the comparability of different sequencing methods for soil microbiome analysis.

Reviewer 2 (Comments for author):	2
Reviewer 2: comment 1:	3
Reviewer 2: comment 2:	4
Reviewer 2: comment 3:	5
Reviewer 2: comment 4:	5
Reviewer 2: comment 5:	6
Reviewer 2: comment 6:	6
Reviewer 2: comment 7:	6
Reviewer 2: comment 8:	6
Reviewer 2: comment 9:	7
Reviewer 2: comment 10:	7
Reviewer 2: comment 11:	7
Reviewer 2: comment 12:	7
Reviewer 2: comment 13:	8
Reviewer 2: comment 14:	8

Reviewer 2 (Comments for author):

This study compares amplicon and shotgun metagenomic sequencing approaches to analyse microbial communities in 131 temperate grassland soil samples from Ireland.

The authors assess how these methods differ in detecting and classifying microbial taxa, their ability to distinguish microbial communities across samples, and their capacity to link microbial communities to nitrogen cycle functions.

The study finds that while both methods broadly identify similar microbial patterns, discrepancies arise mainly due to differences in taxonomic classification methods.

Shotgun sequencing provides deeper taxonomic resolution but requires more computational resources, whereas amplicon sequencing is a cost-effective and widely used alternative.

The findings emphasise the importance of selecting an appropriate sequencing approach based on study objectives and highlight the need for harmonized taxonomic databases for improved comparability of soil microbiome studies.

This study is well-structured and provides a comprehensive comparison of sequencing methodologies used in soil microbial ecology.

The large sample size strengthens the reliability of the findings, and the methodological rigor, particularly the use of multiple reference taxonomies and classifiers, enhances the credibility of the results.

We sincerely thank the reviewer for taking time to review our manuscript and their kind words. The specific queries flagged by the reviewer have been addressed below and within the manuscript.

Reviewer 2: comment 1:

First, the correlation between the relative abundances of phyla detected by shotgun metagenomics and those detected by amplicon sequencing is not clearly presented, making the discussion ambiguous.

While the manuscript states that the results from shotgun and amplicon sequencing are “similar”, this assessment primarily focuses on the presence or absence of phyla rather than evaluating how well the relative abundances of each phylum match between the two methods.

This is a fundamental issue in interpreting the study’s findings.

If the relative abundances between shotgun and amplicon sequencing were consistent, then the correlations between microbial relative abundances and nitrogen cycle-related environmental factors, as discussed in the latter part of the manuscript (L375-386), should also be consistent across both methods.

Conversely, if the relative abundances differ significantly between the two approaches, the premise for discussing correlations with nitrogen cycling is undermined. Therefore, before engaging in discussions on the relationship between microbial communities and nitrogen cycling functions, the manuscript should first establish the correlation between shotgun and amplicon sequencing results.

As it stands, the discussion on nitrogen cycle-related functions is likely not addressing the most critical issue.

We thank the reviewer for bringing this to our attention. As suggested, we conducted a Spearman rank correlation analysis to directly evaluate the comparability of relative abundance estimates between amplicon and shotgun sequencing across the dominant phyla discussed in our study. This analysis revealed that while a subset of phyla, such as Verrucomicrobiota ($\rho = 0.695$, $p < 0.001$) and Actinomycetota ($\rho = 0.525$, $p < 0.001$), exhibited moderate and statistically significant correlations, many others showed weak or no correlation in relative abundance values between methods. These findings are consistent with our Procrustes analysis, which indicated moderate similarity in overall community structure ($R = 0.42$ – 0.43), yet considerable variation at the individual phylum level.

We have now revised the manuscript to clearly distinguish between community-level similarity and individual phylum-level abundance differences. Specifically, we clarify that although the relative abundance values of individual phyla differ across methods, the broader ecological relationships, such as associations between microbial phyla and nitrogen cycling functions, are largely preserved. Accordingly, we interpret these functional associations as qualitatively but not quantitatively consistent across sequencing approaches. This clarification has been incorporated into both the Results and Discussion sections.

Lines 345- 364 in results section:

To assess the consistency of phylum-level abundance estimates between sequencing methods, we performed a Spearman rank correlation analysis across 17 dominant phyla including bacterial, archaeal, and fungal phyla (**Table 1**). Of these, 8 phyla (47%) showed statistically significant correlations in relative abundance between amplicon and shotgun data, including

Verrucomicrobiota ($\rho = 0.695$, $p < 0.001$), Actinomycetota ($\rho = 0.525$, $p < 0.001$), Basidiomycota ($\rho = 0.255$, $p = 0.003$), and Ascomycota ($\rho = 0.223$, $p = 0.0098$). The remaining phyla displayed low correlation coefficients and non-significant p-values, suggesting variable agreement in relative abundance values between methods. These results indicate that while both methods identify similar patterns in phylum prevalence, the relative abundance values are not directly comparable.

Building on the above, we explored whether the ecological relationships between microbial phyla and nitrogen cycling functions (as assessed via potential nitrification, denitrification, and the $N_2O : N_2O + N_2$ ratio) are conserved across sequencing methods, despite differences in relative abundance values.

Lines 449- 455 in discussion section:

Additionally, our Spearman correlation analysis revealed that although nearly half of the dominant phyla exhibited statistically significant correlations between methods, many others did not. This supports our interpretation that community structure is moderately similar, yet the relative abundance values of individual phyla differ. As such, we propose that microbial associations with nitrogen functions should be viewed as qualitatively but not quantitatively consistent when comparing amplicon and shotgun approaches.

Reviewer 2: comment 2:

Second, the manuscript calculates the ratio of microbial groups detected by amplicon sequencing and shotgun metagenomics based on the equation in L227-229, but it does not justify the validity of this approach. This justification is essential because amplicon sequencing targets specific domains (e.g., bacteria and archaea) using domain-specific primers, whereas shotgun metagenomics sequences all DNA present in the sample, including eukaryotic and prokaryotic DNA. As a result, a direct comparison of relative abundances between the two methods may not be appropriate without proper normalization. To ensure the robustness of the analysis, the manuscript should either (i) provide a clear rationale for why this ratio calculation is valid or (ii) discuss the potential biases introduced by this method and consider whether corrections are needed.

Furthermore, the use of Bray-Curtis dissimilarity for MDS (Multidimensional Scaling) analysis (L318-330) may also be affected by this issue. Bray-Curtis dissimilarity is typically applied to non-binary community composition data, and differences in data characteristics between shotgun and amplicon sequencing may introduce biases into the MDS results. The authors should either justify the validity of applying Bray-Curtis dissimilarity to these datasets or consider alternative normalization approaches or distance metrics to ensure that the comparison is not biased by methodological differences between the two sequencing techniques. On the other hand, analyses using rank-based correlations to assess the relationship between microbial communities and nitrogen cycling functions are less likely to be affected by the above concerns. However, to strengthen the study's reliability, it is still crucial to first clarify the extent to which shotgun and amplicon sequencing data are correlated before interpreting their functional associations.

Addressing these points would improve the clarity of the discussion and enhance the robustness of the study's conclusions.

We thank the reviewer for their careful evaluation of the methodological assumptions underlying our comparative approach. Regarding the use of the abundance ratio (shotgun/amplicon), our goal was to identify instances where relative abundance estimates for specific phyla diverged dramatically between methods. The ratio was intended as a qualitative indicator to identify such discrepancies, particularly among dominant phyla. However, we have removed the relative abundance ratio analysis and its associated figures and tables from the manuscript. We agree that direct ratio-based comparisons between amplicon and shotgun sequencing could introduce misleading interpretations due to their differences in domain specificity, sequencing depth, and reference database coverage. This simplification improves the focus and robustness of our comparative framework.

Regarding the use of Bray-Curtis dissimilarity for ordination analysis (PCoA), we acknowledge that differences in sequencing depth, taxonomic resolution, and overall data characteristics between amplicon and shotgun approaches could affect absolute abundance values. However, Bray-Curtis remains a widely accepted metric in microbial ecology for compositional data, particularly when applied to relative abundances and after filtering out rare taxa. In our analysis, we consistently applied relative abundance thresholds (e.g., 0.001%, 0.01%) across both datasets to reduce the impact of low-abundance and inconsistently detected phyla, thereby enabling more reliable comparisons of community composition. To further strengthen our analysis, we complemented Bray-Curtis ordinations with Procrustes analysis, which aligns overall community structure patterns between datasets and is less sensitive to absolute abundance scaling differences. This dual approach provides a conservative and interpretable framework for comparing microbial community composition across sequencing methods.

As noted by the reviewer, we also expanded our analysis by incorporating Spearman rank correlations between amplicon and shotgun-derived relative abundances for dominant phyla (Results: Lines 345- 364). This analysis confirmed moderate correlation for several phyla (e.g., Actinomycetota, Verrucomicrobiota), while others showed little to no correlation. These findings reinforce our interpretation that although abundance values differ, ecological relationships, such as phylum-level associations with nitrogen cycling functions are qualitatively but not quantitatively consistent between sequencing methods.

Reviewer 2: comment 3:

L114: Specify whether this refers to per sample or total across all samples.

We thank the reviewer for making this point. We have addressed this consideration in the L114.

L112-L115: Our research addresses this gap by evaluating shotgun and amplicon sequencing methodologies across 131 diverse soil samples collected across Ireland, with shotgun sequencing yielding an average of 20 million reads per sample.

Reviewer 2: comment 4:

L275-277: Remove “at” before the percentage values to improve readability.

We thank the reviewer for making this point. We have addressed this to improve readability in lines L279-L282.

Lines 279-282: The most abundant bacterial phyla were Pseudomonadota (34.4% +/- 2.6), Actinomycetota (26.4% +/- 4.1), followed by Verrucomicrobiota (4.5% +/- 1.4), Bacteriodota (4.5% +/- 3), and Acidobacteriota (4.1% +/- 1.4). The most abundant archaea were Euryarchaeota (0.7% +/- 0.2) and Nitrososphaerota (0.3% +/- 0.4) followed by 'Candidatus Thermoplasmatota' (0.02% +/- 0.003), 'Candidatus Bathyarchaeota' (0.02% +/- 0.03) and Thermoproteota (0.01% +/- 0.003).

Reviewer 2: comment 5:

L176: Explain why these sequence lengths were chosen.

We thank the reviewer for making the comment. Merged reads with a range between 230 and 413 bp were further filtered to remove chimeras. After denoising, we identified non-target-length sequences by looking at the sequence length distribution. Therefore, we removed sequences that are much longer or shorter than expected. This has been addressed in lines 173 – 179.

Lines 173 – 179: For ITS sequences, no fixed length trimming was applied before denoising to accommodate the natural biological variation in ITS region length. After denoising, we examined the distribution of merged read lengths and retained sequences between 230 and 413 bp, which correspond to the expected size range for fungal ITS2 amplicons. This post-denoising length filtering step was performed to remove potential non-target or poor-quality sequences and improve the accuracy of downstream taxonomic classification.

Reviewer 2: comment 6:

L183: How many cycles of PCR were performed?

We thank the reviewer for making this point. We have added the required information line L184.

L184-L186: One ng DNA per sample was used to prepare the libraries following the manufacturer's instructions which involves 12 cycles of PCR.

Reviewer 2: comment 7:

L200: The reference information appears to be incorrect. Please correct it.

Thank you for your careful review of our manuscript. Regarding the reference in L200, we would like to clarify that the cited study is indeed the correct reference. This study presents our previous in silico analysis, which was further validated on a real dataset, forming the basis for our methodological approach in this manuscript.

Reviewer 2: comment 8:

L306: Should be Table 2.

We thank the reviewer for the comment. We have now removed the said table from the manuscript. Please refer to comment 2.

Reviewer 2: comment 9:

L345: What do categories ii and iii refer to in the status shown in Fig. 3?

We thank the reviewer for bringing this to our attention, we have amended the corresponding paragraph accordingly lines L327- 340.

L327- 340: To this end, we organized phyla into four distinct groups: (i) phyla identified by both methods and sharing the same nomenclature (19 phyla; 23%) (classified as ‘Present in both’ in Fig. 3) ; (ii) phyla detected by both methods but awaiting nomenclature updates in amplicon datasets (7 phyla; 0.1%), such as Proteobacteria now termed Pseudomonadota; (labelled as ‘Phylum renamed’ in Fig. 3) (iii) phyla identified by both methods but subjected to taxonomic reclassification (15 phyla; 18%), for instance, Basidiobolomycota appears as a phylum in the UNITE database but is categorized under the Zoopagomycota phylum in NCBI’s taxonomy (labelled as ‘Taxa reassignment’ in Fig. 3). Similarly, Patescibacteria, identified in SILVA, is treated as a non-ranked group in NCBI, encompassing 57 ‘Candidatus’ phyla; (iv) phyla detected exclusively by one method (18 by amplicon; 21% and 24 by shotgun data; 29%). These are categorized as ‘Absent’ not detected by one method or ‘Present’ detected by only one method in Fig. 3.

Reviewer 2: comment 10:

L408: In cases where the relative abundance obtained from amplicon sequencing is lower than that from shotgun sequencing, primer coverage issues are likely involved. This aspect should also be included in the discussion.

We thank the reviewer for this helpful suggestion. While the limitations of amplicon sequencing, including primer bias, were initially addressed in the Introduction, we agree it is important to reiterate this point in the Discussion. We have now included a sentence explicitly noting the primer bias improving clarity in the Discussion section.

Lines 508-513: While our study provides important insights into the comparability of amplicon and shotgun sequencing, several limitations should be acknowledged. Amplicon sequencing relies on domain-specific primers (specifically V3 region of 16S and ITS region) that may not equally amplify all microbial taxa. As such, underrepresentation of certain groups, particularly those with mismatches to primer regions, is likely attributable to primer bias, a known limitation in marker gene studies (1, 2).

Reviewer 2: comment 11:

L414: References are needed here.

We thank the reviewer for the comment, as the line is referring to our own results we have amended the line to reflect the results we are referring to.

Line 420-422: The comparisons at phylum level revealed moderate similarities between the two methods investigated (Figure 2, Table 1).

Reviewer 2: comment 12:

L470: What does “standardise” mean in this context? Is it referring to nomenclature?

We thank the reviewer for bringing this to our attention. Yes, in this context, “standardise” refers to harmonizing the taxonomic reference databases used for both amplicon and shotgun sequencing to enable more direct comparisons between the datasets. We have now clarified this in the manuscript.

L479-482: While a potential solution could be to harmonise the reference database used across both datasets, the only database currently available for both, Greengenes2, relies on a marker gene approach, which might be inadequate for comprehensive soil shotgun metagenomic analysis.

Reviewer 2: comment 13:

L513: Since this study does not examine sequencing depth, it may be omitted.

We thank the reviewer for the suggestion, this line has been omitted accordingly.

Reviewer 2: comment 14:

L519: Amplicon sequencing can also provide species-level taxonomy.

We thank the reviewer for this important clarification. We agree that species-level identification is theoretically possible with amplicon sequencing, particularly when using full-length 16S rRNA reads or in low-diversity environments. However, in the context of short-read V3–V4 targeted amplicon sequencing, as used in our study, species-level resolution remains limited, especially in high-diversity environments such as soil, where many taxa lack representative reference genomes. In our dataset of 131 soil samples, no taxa could be reliably classified to species level at commonly accepted thresholds ($\geq 97\%$ identity).

This limitation has also been acknowledged in studies of less complex environments, such as the infant gut, where 16S rRNA amplicon sequencing is routinely reported at the genus level due to its higher reliability. A recent systematic review (3) found that species-level insights in such studies were typically reserved for shotgun metagenomics. Furthermore, Schriefer et al. (2018) (4) demonstrated that commonly used 16S pipelines like QIIME can yield high false-positive rates when attempting species-level classification, even in controlled microbiota settings. Taken together, these observations support our decision to focus on phylum-level analyses in the current study.

As the manuscript does not explicitly discuss species-level taxonomy in the context of amplicon data, no changes were made to the manuscript text.

Reviewer 4 (Comments for author):

The revised manuscript is well-written and provides clear insights into the comparability of different sequencing methods for soil microbiome analysis.

We sincerely thank the reviewer 4 for their valuable time and expertise in reviewing our manuscript.

References:

1. Campanaro S, Treu L, Kougias PG, Zhu X, Angelidaki I. Taxonomy of anaerobic digestion microbiome reveals biases associated with the applied high throughput sequencing strategies. *Scientific reports*. 2018;8(1):1926.
2. Jeong J, Yun K, Mun S, Chung W-H, Choi S-Y, Nam Y-d, et al. The effect of taxonomic classification by full-length 16S rRNA sequencing with a synthetic long-read technology. *Scientific reports*. 2021;11(1):1-12.
3. Alcazar CG-M, Paes VM, Shao Y, Oesser C, Miltz A, Lawley TD, et al. The association between early-life gut microbiota and childhood respiratory diseases: a systematic review. *The Lancet Microbe*. 2022;3(11):e867-e80.
4. Schriefer AE, Cliften PF, Hibberd MC, Sawyer C, Brown-Kennerly V, Burcea L, et al. A multi-amplicon 16S rRNA sequencing and analysis method for improved taxonomic profiling of bacterial communities. *Journal of microbiological methods*. 2018;154:6-13.

Re: mSystems01059-24R2 (Consistent microbial insights across sequencing methods in soil studies: The role of reference taxonomies)

Dear Dr. Fiona Brennan:

Your manuscript has been accepted, and I am forwarding it to the ASM production staff for publication. Your paper will first be checked to make sure all elements meet the technical requirements. ASM staff will contact you if anything needs to be revised before copyediting and production can begin. Otherwise, you will be notified when your proofs are ready to be viewed.

Sincerely,
Rachel Poretsky
Editor
mSystems